# Challenges and Opportunities of Universal Health Coverage in Africa: A Scoping Review

**DOI:** 10.3390/ijerph22010086

**Published:** 2025-01-10

**Authors:** Evaline Chepchirchir Langat, Paul Ward, Hailay Gesesew, Lillian Mwanri

**Affiliations:** 1Research Centre for Public Health, Equity and Human Flourishing (PHEHF), Torrens University Australia, Adelaide, SA 5000, Australia; paul.ward@torrens.edu.au (P.W.); hailay.gesesew@torrens.edu.au (H.G.); lillian.mwanri@torrens.edu.au (L.M.); 2Center of Excellence in Women and Child Health East Africa, Aga Khan University, 3rd Parklands Avenue, P.O. Box 30270, Nairobi 00100, Kenya; 3Tigray Health Research Institute, Mekelle 1547, Ethiopia

**Keywords:** universal health coverage, health systems, Africa, health insurance schemes, social health protection schemes, primary health care

## Abstract

Background: Universal health coverage (UHC) is a global priority, with the goal of ensuring that everyone has access to high-quality healthcare without suffering financial hardship. In Africa, most governments have prioritized UHC over the last two decades. Despite this, the transition to UHC in Africa is seen to be sluggish, with certain countries facing inertia. This study sought to examine the progress of UHC-focused health reform implementation in Africa, investigating the approaches utilized, the challenges faced, and potential solutions. Method: Using the Preferred Reporting Items for Systematic Reviews and Meta-Analyses Extension for Scoping Reviews guidelines, we scoped the literature to map out the evidence on UHC adoption, roll out, implementation, challenges, and opportunities in the African countries. Literature searches of the Cochrane database of systematic reviews, PUBMED, EBSCO, Eldis, SCOPUS, CINHAL, TRIP, and Google Scholar were conducted in 2023. Using predefined inclusion criteria, we focused on UHC adoption, rollout, implementation, and challenges and opportunities in African countries. Primary qualitative, quantitative, and mixed-methods evidence was included, as well as original analyses of secondary data. We employed thematic analysis to synthesize the evidence. Results: We found 9633 documents published between May 2005 and December 2023, of which 167 papers were included for analysis. A significant portion of UHC implementation in Africa has focused on establishing social health protection schemes, while others have focused on strengthening primary healthcare systems, and a few have taken integrated approaches. While progress has been made in some areas, considerable obstacles still exist. Financial constraints and supply-side challenges, such as a shortage of healthcare workers, limited infrastructure, and insufficient medical supplies, remain significant barriers to UHC implementation throughout Africa. Some of the promising solutions include boosting public funding for healthcare systems, strengthening public health systems, ensuring equity and inclusion in access to healthcare services, and strengthening governance and community engagement mechanisms. Conclusion: Successful UHC implementation in Africa will require a multifaceted approach. This includes strengthening public health systems in addition to the health insurance schemes and exploring innovative financing mechanisms. Additionally, addressing the challenges of the informal sector, inequity in healthcare access, and ensuring political commitment and community engagement will be crucial in achieving sustainable and comprehensive healthcare coverage for all African citizens.

## 1. Introduction

In African countries, policy reforms and efforts toward health for all date back to 1978, when the Alma Ata Declaration promoted health for all based on the principle of health as a human right [1]. Health for all continues to be a priority for African countries to date. In 2016, the African Union adopted the Africa Health Strategy (2016–2030), which mandates all African governments to guarantee healthcare for all its citizens in an equitable manner by 2030 [2]. This strategy aligns with the 2030 Agenda for Sustainable Development, where commitments to universal health coverage (UHC) are defined [3]. UHC requires that access to quality healthcare is ensured to all who are in need and without those receiving care experiencing financial hardship. The World Health Organization (WHO) provided three dimensions [4] through which countries should focus as they progress towards UHC. These dimensions include (a) expanding priority services, where countries define which services they expand first and why; (b) including more people, where countries describe who to include first and why; and (c) reducing out-of-pocket payments, where countries set out to shift from out-of-pocket payment toward prepayment with clear strategies and rationale.

Studies have shown that on average, it would require a decade to achieve UHC coverage of between 60–90% [5]. In Africa, most UHC reforms across different countries have been in place for more than a decade (Figure 1). However, progress toward UHC dimensions of population, service, and financial coverage has been uneven and slow [6]. In some countries, such as Tunisia, [7] Rwanda, and Morocco [8,9], population and service coverage are between 60–90%, whereas in countries such as Ghana [10], Gabon [11], Nigeria [12], Kenya [13], Zambia [14], and Senegal [15], population coverage and service coverage are between 50–20%.

A number of studies have evaluated Africa’s progress toward universal health coverage and the effectiveness of the reforms in relation to the three dimensions established by the World Health Organization [4,16]. In this review, we set out to extend this debate by focusing on the UHC approaches in Africa, highlighting the UHC reforms adopted in Africa, and highlighting their successes, challenges, and future direction. We answer the following questions: What approaches has Africa taken in advancing the achievement of UHC goals? What challenges are facing these UHC approaches? And what opportunities lie to accelerate progress towards UHC in Africa?

## 2. Method

### 2.1. Study Design

A scoping review was conducted using *The Joanna Briggs Institute Reviewers’ Manual 2015* [17].

### 2.2. Eligibility Criteria

Studies were eligible for inclusion if they (i) were published in the English language between December 2005 and September 2022—the start date of 2005 December was selected since the endorsement of UHC was done during the 58th session of the World Health Assembly in May 2005, where Member States were urged to work towards sustainable health financing for UHC [18]; (ii) those that mention approaches, challenges, and opportunities in one or more of three components of UHC, i.e., service coverage, financial coverage, and population coverage; (iii) those that focused on Africa; and (iv) those that were either primary and secondary qualitative, quantitative, or mixed-methods studies, as well as gray literature. It is worth noting that while the search concluded in September 2022, the review includes studies published up to December 2023, indicating a thorough and up-to-date examination of the most recent developments in the field.

We excluded studies that do not focus on any of the UHC components that were not done in Africa or were not reported in the English language.

### 2.3. Search Strategy

The systematic search was conducted in consultation with a librarian to determine the most acceptable databases for the research question. The search terms used included “Universal health insurance” OR “Universal healthcare” OR “UHC” OR “Universal health coverage” OR “Universal health care” OR “national health insurance”, together with a search filter for African countries. The following databases were searched with an English language restriction: Cochrane Database of Systematic Reviews, PUBMED, EBSCO, Eldis, SCOPUS, CINHAL, TRIP, and Google Scholar. We included citation searches for articles, reports, and editorials by influential commentators and experts on UHC, as well as searched for ongoing or recently completed pertinent systematic reviews on the PROSPERO registry. The searches were primarily conducted using online resources. The final search strategy for all the databases can be found in Appendix A.

### 2.4. Study Selection

The documents identified were entered into the EndNote library and exported to Covidence [19]. Two independent reviewers (E.L. and H.G.) conducted two levels of screening using Covidence. A level one screening, using citation titles and abstracts, was used to determine the study’s relevance to the overall objective of the realist review. A level two screening of full text was used to determine if the citations met the inclusion criteria. The two reviewers independently extracted data with disagreements resolved through discussion. Study findings were extracted using a data extraction form that was initially pilot-tested on three randomly selected included studies before its actual use. E.L. used the form to extract data.

### 2.5. Data Extraction and Synthesis of Results

The following areas were captured on the data extraction form: the citation information, geographical origin, year of publication, study setting, population, objective, methods, design, UHC dimensions implemented, country approach to the UHC dimension implemented, performance of the country’s approach (changes have been observed following the reform), trade-offs made on the path to UHC, lessons learned from the approach, challenges for the present and into the future, unintended effects of the chosen approach, limitations, conclusions, and recommendations. The data were analyzed using thematic analysis [20]. Through an iterative process of coding and thematizing text, the authors were able to draw the UHC approaches implemented in Africa, their underlying challenges, and opportunities from the different Results and Discussion sections of the articles.

## 3. Results

### 3.1. Characteristics of Included Studies

A total of 9633 citations were found, of which 167 papers were eligible for inclusion (Figure 2). The Appendix A (Appendix A) provides the characteristics of the included studies. We removed the duplicates (*n* = 3723), titles, and abstracts that did not match the criteria (*n* = 5315) and full texts that did not meet the inclusion criteria (*n* = 429). Of the 167 studies, 72 (43%) studies originated from multi-country studies, followed by 15 (9%) from Ghana and 12 (7%) from Nigeria.

### 3.2. Universal Health Coverage in Africa: Approaches, Challenges, and Opportunities

In this review we asked these questions: What approaches has Africa taken in advancing the achievement of UHC goals? What challenges are facing these UHC approaches? And what opportunities lie to accelerate progress towards UHC in Africa? The overall findings are organized into three themes: UHC-focused health reforms, challenges facing these reforms, and emerging solutions/opportunities.

#### 3.2.1. Theme 1: UHC Approaches in Africa

Evidence shows that the UHC approaches implemented in Africa have mostly taken two forms, that is, (a) the establishment of social health protection schemes (social health insurance and community-based health insurance) and (b) the strengthening of primary healthcare systems, with a few addressing both (Table 1 and Figure 3). Table 2 shows the comparison of the two UHC approaches in Africa.

(a)Social health protection schemes

Universal health coverage (UHC) is a key component of social protection [21], and social health protection (SHPS) schemes are one of the social protection mechanisms [22]. Social health protection schemes encompass various models, including community-based health insurance (CBHI), social health insurance (SHI), solidarity schemes, and national health insurance (NHI). SHPS can be either contributory, where beneficiaries or employers pay premiums, or non-contributory, where the government covers the costs through taxation. The choice of model depends on the specific context and resources available in each country [23]. In Africa, most countries have taken up SHI/NHI, with a few having taken CBHI as their UHC vehicle of choice.

(i)Social health insurance/national health insurance

Social health insurance, also known as national health insurance (NHI), is a healthcare funding scheme that complements other financing mechanisms such as taxation, private health insurance, and community-based health insurance (CBHI) [24,25]. The primary function of SHI is to pool health risks among participants, providing financial protection against the uncertainty of illness occurrence and the associated treatment costs. SHI schemes are characterized by several key features: mandatory enrollment, often implemented at a national scale, government-provided, and primarily targeting the formal sector [24,25,26]. Historically, the focus only on the formal sector workforce was rooted in public policies that emphasized worker health and labor productivity. However, post-World War II, there has been a shift towards viewing health coverage as a citizenship entitlement and a fundamental right for all [27]. Several African countries, including Gabon, Nigeria, Tanzania, Morocco, Kenya, Egypt, and South Africa [28,29], have implemented social health insurance or national health insurance (SHI) as the approach for UHC (Table 3).

The implementation of social health insurance (SHI) schemes across these countries reveals a complex landscape of progress and challenges. While each country has adopted unique approaches, several common themes emerge: (a) Countries have innovated with tax levies on specific sectors, while others, like Ghana, rely heavily on VAT-based contributions. The sustainability and adequacy of these funding models remain a critical concern; (b) most countries have struggled to achieve comprehensive population coverage, with informal sector workers and the poor often left behind; and (c) the scope and uniformity of benefit packages vary significantly. While some countries offer uniform benefits, others have fragmented packages that may not adequately address population health needs. In addition to these, common issues such as inadequate infrastructure, staff training deficits, delayed reimbursements, and quality of care concerns are particularly pronounced. Financial pressures, stakeholder conflicts, and political resistance also play a big role and have often led to implementation delays.

(ii)Community-based health insurance

Community-based health insurance (CBHI) schemes, also known as mutual health organizations or micro-insurance plans, represent a grassroots approach to healthcare financing. These initiatives are designed by and for local communities, pooling resources and overseeing the management of the scheme. CBHI was primarily introduced to serve the informal sector, which comprises 70–90% of the population in many African countries [5]. These schemes typically feature equal premiums for all participants and are often voluntary in nature, with Rwanda being a notable exception where enrollment is mandatory [44].

Rwanda’s CBHI scheme stands out as a success story in sub-Saharan Africa, being the only country where more than 90% of the population is covered by a community-based health program. The mandatory nature of the scheme and significant donor support have contributed to its success [45].

Tanzania’s journey with CBHI has been marked by continuous efforts to improve and adapt the system. In 1996, CBHI made its entry as an initiative to make healthcare services available and affordable to the people residing in rural areas and those in the informal sector. In 2001, the government enacted the Community Health Fund (CHF) Act and declared CHF a voluntary prepayment health financing mechanism to be rolled out countrywide. In 2009, the government established Tiba Kwa Kadi (TIKA) urban areas, and CHF was merged with the national health insurance fund to address challenges and improve coverage [46].

Senegal’s government has made significant progress towards UHC, with the insurance coverage increasing from 20% in 2012 to 49.6% in 2018, with CBHI accounting for about 20% of the total insured population. It has also established community-based organizations across the country in addition to providing subsidies, including 100% for the elderly and disabled [15,47].

Burkina Faso, on the other hand, has struggled with low insurance coverage, with only 0.5% of women and 1.5% of men covered by 2010. In 2015, a bill aimed at introducing income-based premiums was put forth, but progress has been slow due to financial and governance challenges [48].

In Uganda, health insurance coverage is equally low at 7.6 despite its CBHI making its first entry in the year 1995. The scheme, despite being open to all, sees the poor and vulnerable often excluded due to the inability to pay premiums. Low enrollment is also attributed to limited accredited health facilities, which could be a result of a shift in focus by the government from social policies to economic growth and stagnating public health expenditure [49,50].

In Ethiopia, the CBHI is characterized by strong government involvement, with community participation in scheme design, management, and supervision [51].

In the Democratic Republic of Congo (DRC), the UHC journey started in 2009, choosing community health insurance as the primary vehicle for achieving this goal. Despite this initiative, the DRC faces numerous obstacles in implementing effective CBHI schemes and achieving its health targets [52].

While Rwanda and Senegal have shown promising growth, the other countries still face considerable challenges in extending coverage and providing equitable access to healthcare services.

(b)Strengthening primary healthcare for UHC

An alternative or complementary approach to achieving universal health coverage is strengthening primary healthcare (PHC) systems. Bayarsaikhan and colleagues (2022) argue that robust health systems, financing, and service delivery platforms are crucial for providing equitable access to quality services and adequate financial protection. Focusing on PHC is considered one of the most cost-effective ways to increase service coverage for UHC. Strengthening PHC involves several key elements, such as the construction of additional primary healthcare facilities, recruitment of more staff to run the facilities, ensuring access to medicines and technologies to diagnose and treat medical conditions and prevent disability, and training the human resources for health to serve primary healthare facilities [53,54].

Several African countries have recognized the importance of primary healthcare systems as the foundation for achieving universal health coverage. These nations have implemented strategies that go beyond insurance schemes to address the broader aspects of health system strengthening: Some notable examples include: Mali [55] focusing on community-based primary healthcare and health worker training; Tunisia [7], implementing a comprehensive primary healthcare network, with a focus on preventive services; South Sudan [56], prioritizing the development of basic health infrastructure in rural areas; Zambia [14], expanding primary healthcare facilities and improving supply chains for essential medicines; Ethiopia [57], deploying health extension workers to improve access to basic health services in rural areas; Cameroon [58], depends on its public health system, though with challenges; South Africa [28], implementing a health insurance program with a strong emphasis on primary healthcare; Sierra Leone [59] and Malawi [60] launched a nationwide free healthcare initiative (FHCI) within the public healthcare system; and Seychelles [61], a high performer in terms of overall service readiness and availability in the African healthcare landscape, through a publicly funded, organized, and owned health system.

#### 3.2.2. Theme 2: Challenges Facing UHC Approaches in Africa

While African nations have chosen varying approaches as their vehicles to progress toward UHC, each approach of the approaches offers unique benefits and faces distinct challenges in the African context.

For countries that have taken up CBHI, the schemes are seen to offer certain advantages, such as being suitable for countries that have no resources to set up and administer a national health insurance plan. Evidence from these countries, however, suggests that this approach may not be a viable long-term solution for achieving UHC as several key limitations exist. They include the inability to mobilize sufficient amounts of funds; the inability of the scheme to provide comprehensive coverage due to its low purchasing power; a high fragmentation of the scheme that leads to small risk pools and increased administrative costs; limited benefits to beneficiaries, as they are often restricted to their community/boundary of enrollment and presence of co-payments for inpatient hospital care, which still creates financial barriers to accessing care; and the voluntary nature of the scheme leads to adverse selection and low enrollment among the poor and vulnerable populations, hence having little effect on access to care for these target groups [4,15,24,26,52,62,63]. These limitations highlight the need for more comprehensive and sustainable approaches to achieving UHC in Africa, potentially through the integration of CBHI with broader national health insurance programs or the strengthening of public healthcare systems.

On the other hand, SHI/NHI has been implemented in Africa but faces significant challenges despite showing huge success in high-income countries with large formal sectors, such as Germany and the Netherlands [64]. One primary obstacle is the predominance of the informal sector in many African economies, accounting for 70–90% of the workforce in many countries [65]. This large informal sector presents several difficulties for SHI implementation, such as limited revenue collection from informal workers, difficulty in enforcing mandatory enrollment, challenges in accurately assessing income for premium calculations, and increased risk of adverse selection. This issue is compounded by fragmented benefit packages, which lead to inconsistencies in coverage across different schemes. Additionally, many African countries face operational challenges, including inadequate infrastructure, insufficient staff training, and limited financial resources [31,35,37,39,42,66]. Despite these challenges, the majority of African countries have adopted or are slowly moving to SHI/NHI as their preferred approach to achieving universal health coverage. This necessitates innovative strategies to overcome these obstacles if SHI/NHI is to deliver UHC for African countries.

While some countries have implemented diverse approaches to strengthening the primary healthcare system for UHC, each country faces unique challenges. Some of the common themes that emerge include the critical role of political will to support the healthcare system, the need for sustainable financing mechanisms, the importance of addressing human resource challenges, and the value of community engagement in healthcare delivery [7,14,28,55,56,58,59,61].

#### 3.2.3. Theme 3: Emerging Solutions/Opportunities for Accelerating UHC Progress in Africa

To address the challenges facing UHC implementation in Africa, some solutions have been proposed for accelerating progress towards UHC in Africa; they include the following: strengthening health system building blocks to improve overall healthcare delivery [67]; improving tax collection efficiency to increase available funds for healthcare [68,69]; allocating more funds to the health sector from national budgets [68]; exploring new revenue sources to diversify healthcare financing [69,70]; implementing effective risk pooling mechanisms to spread financial risk [53,71,72]; adopting strategic purchasing practices to optimize resource allocation [73,74]; and reducing insurance administrative costs and inefficiencies to maximize available resources [15,68,75,76]. In addition to this, political goodwill [77,78] and buy-in from the community [15,51,79] are crucial for driving and sustaining UHC initiatives. By addressing these critical success factors, African countries can enhance their chances of achieving meaningful progress towards UHC.

## 4. Discussion

The World Health Organization (WHO) has been instrumental in guiding Member States towards universal health coverage. In 2005, the World Health Assembly Resolution 58.33 recommended restructuring healthcare financing to prioritize prepayment methods. This approach aims to reduce out-of-pocket expenses and increase access to essential health services [80]. African countries have made significant strides in embracing UHC principles. By 2020, an impressive 93% of African nations reported progress in their commitment to UHC [6]. Many countries have developed policies specifically addressing UHC, while others have incorporated UHC goals into their broader health sector strategies. This widespread adoption demonstrates a growing recognition of UHC’s importance in improving population health and reducing financial hardship associated with healthcare costs.

From this review, we see that African countries are focusing increasingly on the financial protection dimension of UHC, where the implementation of health insurance as a prepayment funding method to reduce out-of-pocket (OOP) payments for healthcare services is dominant. Evidence from the review suggests that social health protection schemes do improve the use of health services; however, their effects on financial protection and quality of care are mixed [81]. For countries implementing CBHIs, for example, we found that the scheme provides financial protection by reducing OOP spending and increasing access to healthcare, as seen by the increased rates of utilization of care. However, there was also a strong indication that they still exclude the poorest and perhaps those most in need. This finding is consistent with other reviews where, despite CBHI being lauded for being able to resource and mobilize faster and suitable for countries that have no resources to set up and administer a national health insurance plan, they are deemed not a viable option for delivering UHC in Africa [4,82]. For those implementing social health insurance/national health insurance (SCH/NHI) schemes, it provided more comprehensive coverage and more equitable access to healthcare services. However, they face several challenges in implementation and effectiveness. It inhibits access to care among those unable to pay, particularly the poor and disadvantaged populations, and exhibits long and complex registration requirements and processes; is insufficiently funded; has unaffordable premiums; and the health provider factors, such as availability and accessibility of healthcare facilities, continue to contribute to low uptake and stagnation of the SHIs/NHIs in Africa [68,83].

While many African countries have focused on establishing insurance schemes, several African countries have also adopted primary healthcare-strengthening approaches to achieving universal health coverage [7,14,28,55,56,58,59,61]. These countries have implemented various strategies to enhance their PHC systems, such as increasing the number of community health workers, improving rural health facilities, and integrating PHC services with broader health initiatives.

Overall, some of the possible opportunities for accelerating progress towards UHC in Africa from our analysis of the literature in the scoping review include the importance of long-term planning and gradual system development [7,56], the need for consistent political and financial support [63], the importance of addressing fundamental health system issues before implementing UHC [14,57,58], the impact of political changes on healthcare policies and the ongoing struggle to overcome historical inequities [28], and the effectiveness of strong government commitment [51,56,67,84] and bringing healthcare closer to rural populations [55,57].

### Limitations

This review acknowledges a significant limitation in its inability to access full texts of studies not published in English. This challenge highlights the linguistic diversity prevalent across the African continent, where research may be published in languages such as French, Arabic, Portuguese, or various indigenous languages. The exclusion of non-English studies potentially creates a bias in the review’s findings, as it may overlook valuable insights from Francophone, Lusophone, or Arabic-speaking African countries. This limitation underscores the need for more inclusive research methodologies and increased efforts in translating and disseminating research across language barriers to ensure a truly comprehensive understanding of UHC reforms in Africa. This study also acknowledges that the last question of the review, “What opportunities lie to accelerate progress towards UHC in Africa?” may not be fully and exhaustively addressed using a scoping review method only; however, from this scoping review, we provide some of the possible strategies to accelerate progress towards UHC in Africa. In addition, the findings are current only up to December 2023. This cutoff date is crucial for understanding the context of the review’s conclusions and recommendations. Healthcare policies and reforms can evolve rapidly, especially in response to global events, technological advancements, or shifts in political landscapes. Researchers and policymakers using this review should be aware that significant developments may have occurred since the cutoff date.

Despite these limitations, the comprehensive nature of this review provides a solid foundation for evidence-based policymaking in African countries. Policymakers can draw upon the synthesized findings to refine their approaches, learning from past experiences and adapting to their unique contexts. The inclusion of studies from various African countries also facilitates cross-country learning and comparison. Best practices and successful strategies identified in one context can be adapted and applied in others, promoting regional cooperation in advancing UHC.

## 5. Conclusions and Future Direction

Despite the difficulties of implementing universal health coverage throughout Africa, there is a growing acknowledgment of its usefulness in improving access to health care, evidenced by the increase in population coverage. This could include strengthening public health systems in addition to health insurance schemes to create a comprehensive UHC strategy, looking into innovative financing mechanisms and improving efficiency in existing systems to ensure the long-term sustainability of UHC programs, developing mechanisms to ensure the poorest and most vulnerable populations are not left behind, and fostering partnerships between the public and private sectors.

To move forward, successful UHC implementation in Africa would probably require a multifaceted approach. In addition, addressing the issues of the informal sector, as well as guaranteeing political commitment and community engagement, will be critical to attaining long-term and comprehensive healthcare coverage for all Africans.

## Figures and Tables

**Figure 1 ijerph-22-00086-f001:**
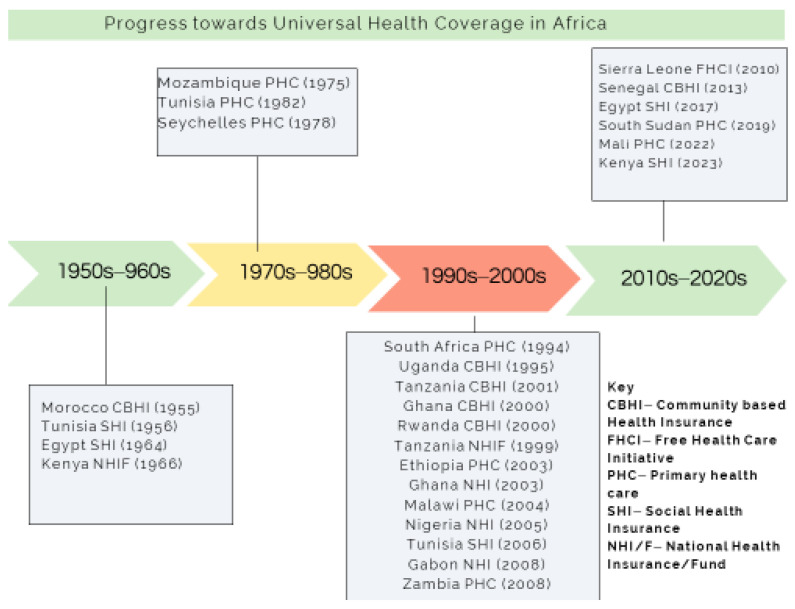
Chronology of progress towards universal health coverage in Africa.

**Figure 2 ijerph-22-00086-f002:**
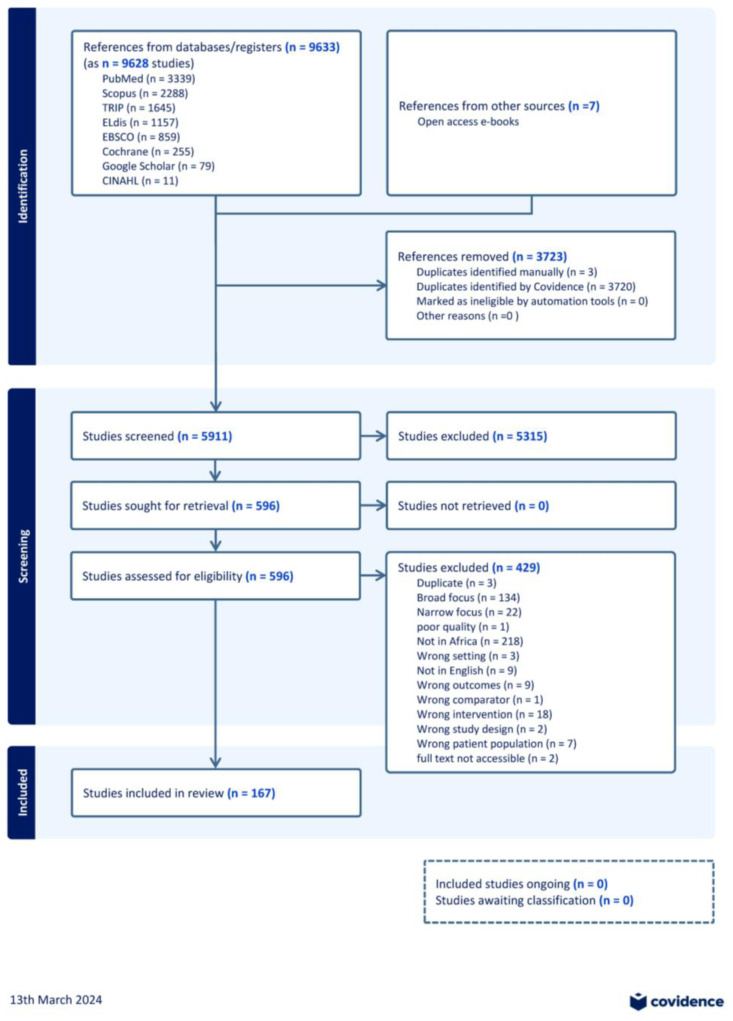
PRISMA flow chart.

**Figure 3 ijerph-22-00086-f003:**
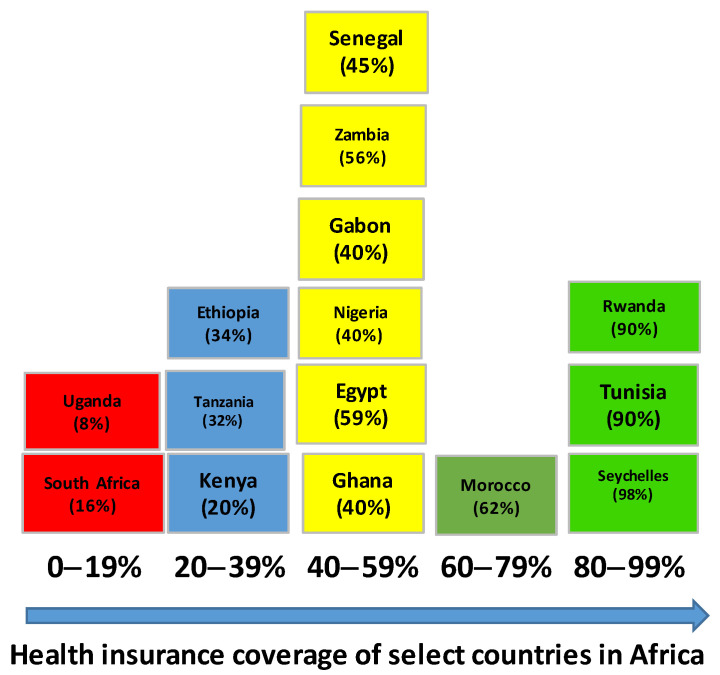
Health insurance coverage in select countries in Africa as of 2023.

**Table 1 ijerph-22-00086-t001:** Universal Health Coverage (UHC) approaches implemented in Africa.

Country	Years Different Since Initiating Different UHC Reforms	UHC Approach
Morocco	1955	CBHI
Tunisia	1956	SHI
Egypt	1964	SHI
Kenya	1966	NHIF
Mozambique	1975	PHC
Seychelles	1978	PHC
Tunisia	1982	PHC
South Africa	1994	PHC
Uganda	1995	CBHI
Tanzania	1996	NHIF
Rwanda	2000	CBHI
Ghana	2000	CBHI
Tanzania	2001	CBHI,
Ghana	2003	NHI
Ethiopia	2003	CBHI/PHC
Malawi	2004	PHC
Nigeria	2005	NHI
Morocco	2005	SHI
Tunisia	2006	SHI
Gabon	2008	NHI
Zambia	2008	PHC
Sierra Leone	2010	FHCI
Senegal	2013	CBHI
Egypt	2017	SHI
South Sudan	2019	PHC
Mali	2022	PHC
Kenya	2023	SHI

**Table 2 ijerph-22-00086-t002:** Comparison of SHI, CBHI, and PHC UHC approaches in Africa.

Aspect	Social Health Insurance (SHI)	Community-Based Health Insurance (CBHI)	Primary Healthcare (PHC)
Focus	Financial risk pooling provides comprehensive coverage and reduces out-of-pocket expenses for healthcare services	Provide financial risk protection, pool resources, and expand access to healthcare services.	Strengthening health systems at the local level, service delivery and access
Implementation	National-level schemes	Local and community-based	National and community-based
Challenges in Africa	Large informal sector, high administrative costs, inequitable contribution schemes, and challenges in ensuring access and utilization for all.	Small risk pools, coverage gaps, limited access for marginalized populations, and insufficient funding	Infrastructure and resource constraints
Equity	May exclude informal workers	Aims for universal access within the community context	Aims for universal access
Cost-effectiveness	Depends on the formal sector size	Generally cost-effective but limited by resource constraints	Highly cost-effective, emphasizing prevention and basic care

The SHI, CBHI, and PHC approaches reveal unique strengths and challenges in the African context. Rather than viewing these approaches as mutually exclusive, policymakers and health professionals should consider how they can be integrated to create comprehensive and effective health financing systems.

**Table 3 ijerph-22-00086-t003:** Social/national health insurance UHC approaches in Africa.

Country	Scheme	Year Established	Implementation
Nigeria	The National Health Insurance Scheme (NHIS)	Established in 1999 under Act 35 of the Constitution [12].	Became operational in 2005 as the primary vehicle for UHC [12].The scheme has specific programs for different segments of the society. Despite its early inception, progress has been slow. As of 2022, the population coverage had reached only 39% [30]. Operationalization of the NHIS in terms of infrastructure for data monitoring, adequate staff with training on health insurance management, adequate finances, and fragmentation of benefit packages continue to challenge NHIS implementation [5].
Ghana	The National Health Insurance Scheme (NHIS)	Established in 2003 through government legislation [31].	It is funded predominantly through tax revenue, contributing 74% through the national health insurance levy (NHIL) and a 2.5% levy on goods and services collected under the value-added tax (VAT). Ghana’s NHIS is mandatory, and the benefit package is uniform for all members, regardless of their contribution levels or sector affiliation [31]. The population covered under the NHIS in Ghana reached 40% within a decade of its implementation and has since stagnated at that level for more than 10 years. Some of the challenges identified for its stagnation are implicit rationing, whereby despite the NHIS starting to operate a generous benefits package covering about 90–95% of common diseases, the issues around a shortage of equipment and medical supplies have restricted access to these services, serving as a major cause of dissatisfaction and disinterest in joining the scheme by the population [31].
Gabon	The National Health Insurance Scheme (NHIS)	Established in 2008.	Mainly funded by a government tax levy on mobile phone companies and on money-sending services, and through contributions from formal and private sector workers. By 2011, Gabon was close to reaching the entire population of 1.5 million people, with evidence indicating an increase in service utilization in all hospitals [11].
Tanzania	The National Health Insurance Fund (NHIF)	Established by Act of Parliament No. 8 of 1999 and began operations in June 2001 [32].	Initially intended for public sector employees, but later expanded to allow voluntary enrollment from other sectors. By 2018, 32% of the Tanzanian population had some form of health insurance, with 8% under NHIF, 21% under community-based health insurance (CBHI), and 3% under private schemes [33,34]. Despite there being a steady increase in coverage by NHIF from 2.0% in 2001 to 8% in 2018, the increase is said to be very slow. Certain elements emerged as to what may have influenced the low enrollment. This included the low involvement of the community in the planning and implementation of the scheme, the premiums rates not being commensurate to the community’s ability to pay, long waiting times at health facilities compared to cash-paying clients, unavailability of some services provided through NHIF, and poor mechanism of reimbursement of NHIF bills to NHIF-accredited health providers affecting the quality of services.
Morocco	Social Health Insurance (SHI)	Established in 2005 [8].	The first scheme, introduced in 2005, is compulsory medical insurance for formal employees in both private and public sectors (AMO), covering 34% of the population. The second scheme, Régime d’Assistance Médicale (RAMED), launched in 2013 for the poor and vulnerable, increased population coverage to 62%. The third scheme, introduced in June 2017, caters to the self-employed [8]. Despite the existence of these schemes, collective health financing is still limited as households cover more than half of total health expenditure out of pocket. Also, the design of the scheme is that they operate independently of each other, leading to fragmentation of the pooling of resources and increasing the administrative cost [35]. As such, recommendations to move towards progressive convergence, revising the benefits package in different ways, and building strong partnerships with the private sector through contracting were made.
Kenya	The National Health Insurance Fund (NHIF)	Established in 1966 [36].	Launched in 1966, it initially targeted employed workers and was limited to inpatient care. In 2018, UHC was declared one of the country’s ‘Big 4’ agendas, leading to the passing of the UHC policy in 2020, with NHIF being proposed as the vehicle to deliver UHC [36]. Currently, it covers 20% of the population and operates three main schemes: The Civil Service Scheme (CSS), the National Scheme (Supa Cover), and the Health Insurance Subsidy for the Poor (HISP). Each scheme offers different benefit packages, with considerable variation between inpatient and outpatient care [37,38]. In 2015, NHIF introduced several reforms to accelerate progress toward UHC, including revised premium contribution rates, expanded benefit packages, and new provider payment methods [39].
Egypt	Social Health Insurance (SHI)	Established in 1964 [40].	The roots of UHC in Egypt can be traced back to the 1950s and early 1960s, when free healthcare was declared for all citizens. However, financial pressures and resource shortages led to a shift towards a mixed system of fee-for-service and limited cost-recovery approaches by 1964. The same year saw the implementation of the national health insurance organization, which initially covered public and private sector employees, pensioners, and widows. Reform initiatives between 2005 and 2015 failed due to conflicts over goals and political processes. In December 2017, the Egyptian parliament passed a bill mandating universal health insurance for all citizens, aiming to increase coverage from the existing 58% of the insured population. Despite 58% insurance coverage by the mid-2000s, out-of-pocket expenses have remained high at 72% [40].
South Africa	National Health Insurance (NHI)	Proposed bill was approved in June 2023 [41].	The 1944 Gluckman Commission in South Africa proposed a fully tax-funded national health service (NHS) to provide free healthcare at the point of service through the establishment of primary healthcare centers [42]. The Gluckman Commission’s proposals, however, were never implemented. In the following decades, South Africa experienced a push towards privatization of health services, leading to the establishment of private health insurance schemes. These schemes, however, only covered a small portion of the population, leaving a significant gap in healthcare coverage. In addition, numerous government-established policy committees were put in place to look into a national health insurance (NHI) system and financing for health, but proposals remained opposed, mainly by the national treasury [42,43]. Despite more than a decade of policy recommendations for the implementation of NHI, South Africa’s National Assembly finally approved a landmark proposed bill in June 2023 that will pave the way for universal health insurance for all South Africans [41].

## Data Availability

The raw data supporting the conclusions of this article will be made available by the authors upon request.

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
