# Peer review of "Challenges and Opportunities of Universal Health Coverage in Africa: A Scoping Review"

_ijerph, 2025, doi:10.3390/ijerph22010086_

Round 1
Reviewer 1 Report
Comments and Suggestions for Authors
Dear Authors,
Following the review procedure of the manuscript, it is recommended that the title be revised. The suggested title is as follows: The title should be changed to:
"A Discussion of the Challenges and Opportunities of Universal Health Coverage in Africa..." It is challenging to ascertain the success of UHC reforms, given the lack of implementation, as evidenced by the case of South Africa. Universal coverage is not aligned with the provision of private insurance for those with financial support. I am of the opinion that this is the case. Furthermore, the research strategy yielded evidence up to 2020 (e.g. Kenya) and extended to 2023 for other countries (e.g. South Africa). Some inconsistencies in the methodology require clarification. The research strategy lacks clarity, and the selection of articles is not transparent.
The objective of the scoping review is to identify both published and unpublished evidence. Grey literature comprises unpublished material, including reports published by the World Health Organisation and books, as well as information from websites of international organisations. Please clarify the discrepancies in the manuscript.
The criteria should be clearly delineated, specifying both the inclusion and exclusion criteria. It is imperative that the authors distinguish between the criteria for inclusion and exclusion. It would be beneficial to consider extending the research period beyond 2022, perhaps to December 2023. The authors should provide clarification on how they will determine the relevance of the systematic reviews on the PROSPERO registry. The results section requires further elaboration, and Table 2 should be enhanced to include the selected articles. The discussion section is challenging to review due to the lack of clarity regarding the research findings.
Comments on the Quality of English LanguageThe English could be improved to more clearly express the research.
Reviewer 2 Report
Comments and Suggestions for Authors
This manuscript sought to describe literature on UHC adoption, roll out, implementation, challenges and opportunities in the African countries.
I thoroughly enjoyed reading this manuscript. It gave a very comprehensive overview of the pathway to UHC using the 3 dimensions (expanding priority services, including more people, and reducing out of pocket costs) as a way of evaluating the progress of each country. Scoping reviews, in general, are wordy but Table 2 could be reorganized to make it less text heavy (maybe something more similar to Table 3).
Reviewer 3 Report
Comments and Suggestions for Authors
Comment.
Studying is very interesting and relevant to the field, and it is therefore important to establish how it has developed and the challenges that lie ahead in relation to UHC.
The Scoping Review method is used to determine the approaches adopted in relation to universal health coverage in different African countries, for which the following questions are raised:
What approaches has Africa adopted to advance in achieving the objectives of UHC?
What challenges do these approaches to UHC face?
What opportunities exist to accelerate progress towards UHC in Africa?
The data were analyzed using thematic analysis. Deductive and inductive coding was carried out to construct the themes on the approaches to UHC in Africa. To establish what limitations the thematic analysis has, since its approach is very broad or flexible if it is not adequately adjusted to the research question, it could compromise the analysis of the studies. This leads me to ask how the methodological strategy was triangulated or used to combine the different studies involved, systematic reviews, meta-analyses, qualitative studies (different study designs) among others.
Although the method allows covering many topics and areas, it would be advisable to establish or describe the strategy to establish the approaches sought and the results. (How the conclusion was reached) It is not found in the study and possibly is not part of its objective, to compare the information on the UHC approaches with problems of population growth or decline, economic variables, how much of the GDP per capita is allocated to health? How is the system being financed. This is relevant to be able to affirm what is said later in the conclusions
The conclusions state, “Despite the difficulties presented by the implementation of universal health coverage throughout Africa, its usefulness in improving the health of the population and eliminating disparities in health care is increasingly recognized” This is a comment from the researchers or arises from the research. How this first conclusion was reached. The important thing in this first part was to define which approaches are the prevailing ones.
Subsequently, the difficulties that are adequately addressed in the second part “This could include strengthening public health systems in addition to health insurance schemes to create a comprehensive universal health strategy, studying innovative financing mechanisms and improving the efficiency of existing systems to ensure the long-term sustainability of universal health coverage programs, developing mechanisms to ensure that the poorest and most vulnerable populations are not left behind and fostering partnerships between the public and private sectors”. This would be a recommendation, “to move forward, successful implementation of universal health coverage in Africa will likely require a multifaceted approach”. “In addition, addressing issues of the informal sector, as well as ensuring political commitment and community participation, will be essential to achieving comprehensive and long-term health coverage for all Africans”.
The last question, not clearly answered, is what opportunities exist to accelerate progress towards UHC in Africa? Therefore, it is important to limit the question posed and what each of the publications expresses. An important limitation is the diversity of publications with different methods, which, although exhaustive, requires a strategy to understand some of the statements made in the manuscript.
